# Feature Learning and Random Features in Standard Finite-Width Convolutional Neural Networks: An Empirical Study

Maxim Samarin[1]             Volker Roth[1]             David Belius[1]

[1]Department of Mathematics and Computer Science, University of Basel, Switzerland

## Abstract

The Neural Tangent Kernel is an important milestone in the ongoing effort to build a theory for deep learning. Its prediction that sufficiently wide neural networks behave as kernel methods, or equivalently as random feature models arising from linearized networks, has been confirmed empirically for certain wide architectures. In this paper, we compare the performance of two common finite-width convolutional neural networks, LeNet and AlexNet, to their linearizations on common benchmark datasets like MNIST and modified versions of it, CIFAR-10 and an ImageNet subset. We demonstrate empirically that finite-width neural networks, generally, greatly outperform the finite-width linearization of these architectures. When increasing the problem difficulty of the classification task, we observe a larger gap which is in line with common intuition that finite-width neural networks perform feature learning which finite-width linearizations cannot. At the same time, finite-width linearizations improve dramatically with width, approaching the behavior of the wider standard networks which in turn perform slightly better than their standard width counterparts. Therefore, it appears that feature learning for non-wide standard networks is important but becomes less significant with increasing width. We furthermore identify cases where both standard and linearized networks match in performance, in agreement with NTK theory, and a case where a wide linearization outperforms its standard width counterpart.

## 1 INTRODUCTION

The Neural Tangent Kernel (NTK) [Jacot et al., 2018] is a seminal contribution to the study of deep neural networks which extended important insights about the connection of Gaussian processes and neural networks [Neal, 1996, Williams, 1996, de G. Matthews et al., 2018, Lee et al., 2018, Garriga-Alonso et al., 2019]. Ever since, subsequent investigations have refined our view on the NTK with results suggesting both its validity as well as insufficiency as an explanation for the performance of practical finite-width neural networks, and the focus of investigation has moved to the network parameterization and architecture differences and their relationship to the NTK [Chizat et al., 2019, Lee et al., 2019, Chen et al., 2020, Hanin and Nica, 2020, Xiao et al., 2020, Seleznova and Kutyniok, 2022]. The NTK framework has inspired work in many directions like infinite ensembles of trees [Kanoh and Sugiyama, 2022], federated learning [Huang et al., 2021] and thus continues to stimulate various advances in deep learning theory.

Jacot et al. [2018] proved that when modeling neural network training under gradient flow, i.e. full batch gradient descent of infinitesimal step size, the training trajectory $f(w, x)$ satisfies an ordinary differential equation (ODE) involving the finite-width Neural Tangent Kernel

$$\Theta(w, x, x') = \left\langle \nabla_w f(w, x), \nabla_w f(w, x') \right\rangle$$
$$= \sum_{i=1}^{p} \frac{\partial}{\partial w_i} f(w, x) \frac{\partial}{\partial w_i} f(w, x') \qquad (1)$$

for weights $w \in \mathbb{R}^p$ and inputs $x, x' \in \mathbb{R}^d$. The form of this kernel depends on the network architecture and the current time-dependent weights $w$ as well as a particular initialization. In this *NTK parameterization*, they showed that when scaling the learning rate per layer in an appropriate way and letting the width tend to infinity, the kernel converges to the infinite-width NTK $\hat{\Theta}$ which is *independent* of the weights and *stays constant* during training, greatly simplifying the ODE in this limit. They showed for the $L^2$ loss that the predictor at convergence is precisely what a kernel regression using the infinite-width NTK would produce. Importantly, the infinite-width NTK depends only on the architecture of the network; it is not learned and thus data-independent.

*Accepted for the 38th Conference on Uncertainty in Artificial Intelligence* (UAI 2022).

The formalism was extended from fully-connected to other architectures including convolutional networks [Arora et al., 2019, Yang, 2019], recurrent neural networks [Alemohammad et al., 2020], residual networks [Huang et al., 2020] and more general architectures [Yang, 2020].

This result can be understood as the convergence of wide networks to random feature models [Chizat et al., 2019]. Let $f : \mathbb{R}^d \to \mathbb{R}^L$ be the function given by a network parameterized by weights $w \in \mathbb{R}^p$ with input $x \in \mathbb{R}^d$ and let $f^l$ be the output in component $l = 1, \ldots, L$, with $L$ being typically the number of classes in a classification task. For $w$ sufficiently close to the random initial weights $w_0$ and $u = w - w_0$, the first order Taylor expansion in the weights

$$f^l(w, x) \approx f^l(w_0, x) + \nabla_w f^l(w_0, x) \cdot u =: f^l_{\text{lin}}(u, x) \quad (2)$$

is an accurate approximation. The right-hand side $f_{\text{lin}}(u, x)$ is a random feature model with weights $u \in \mathbb{R}^p$ and the feature mapping $\phi(x) \in \mathbb{R}^{L \times p}$ is given by the gradients $\phi^l(x) = \nabla_w f^l(w_0, x)$ with respect to the weights at initialization $w_0$. If approximation (2) holds, then also the gradients $\nabla_w f(w, x)$ and $\nabla_u f_{\text{lin}}(u, x)$ of the two models will be close. When training these models with some form of gradient descent and sufficiently small step size for a sufficiently small number of steps, then the training trajectories will stay close, as long as the weight vectors remain in the region around $u = 0$ or $w = w_0$, respectively. Using an $L^2$ loss in over-parameterized models, one can expect both models to converge to zero loss [Du et al., 2019]. If convergence occurs before leaving this region, then the models – whether trained with early stopping or until convergence – will predict a similar function. For the infinite-width case, Lee et al. [2019] proved that $f(w, x)$ and $f_{\text{lin}}(u, x)$ converge in distribution to the same Gaussian distribution. Furthermore, the NTK result can be proved by showing that for very wide neural networks the models $f(w, x)$ and $f_{\text{lin}}(u, x)$ reach zero loss and thus stop evolving before leaving the region where the approximation in Eq. (2) is accurate [Chizat et al., 2019, Lee et al., 2019], known as *lazy training*. The linearized model $f_{\text{lin}}$ does not learn a representation but uses the random representation $\nabla_w f^l(w_0, x)$ which is fixed by the initial weights $w_0$ and remains unchanged throughout training. More in line with Gaussian processes and random feature models [Rahimi and Recht, 2007] but at odds with general intuition on deep learning, NTK theory predicts that, at large widths, a network and its linearization behave similarly and no significant feature learning takes place. This seems to imply that – even for standard neural networks – *learning* a good representation might become decreasingly relevant with increasing over-parameterization.

Motivated by this conjecture, we study standard convolutional neural networks (CNNs) and their respective linearizations (at initialization) given by Eq. (2). We complement previous work in that direction (see Sec. 2) and extend these results for more standard architectures in more common classification tasks. In particular, we perform a thorough study of two standard CNNs, LeNet [LeCun et al., 1998] and AlexNet [Krizhevsky et al., 2012], for increasingly difficult classification tasks at different widths. We observe test accuracy gaps between these networks, in line with the idea that standard neural networks perform feature learning while their linearizations do not. For the wider networks the generalization gap closes, in line with NTK theory, supporting the picture summarized in Fig. 1.

However, we also observe low training accuracy for the linearized networks. We investigate numerical issues which can explain reduced training performance in the linearizations and consider a simplified binary classification setting in which we can solve the linear system in Eq. (2) with a standard solver achieving 100% training accuracy, but observe that this generally causes even worse test accuracy for the linearized models.

In this work, we make the following contributions:

- We show that for (nearly) all considered widths, there is a prominent performance gap between the standard and linearized LeNet and AlexNet and this gap increases when the classification task increases in difficulty. This is shown for MNIST, CIFAR-10, and a subset of ImageNet. We believe this gap exhibits the importance of feature learning for non-wide standard networks.

- We present further instances where wide linearized networks perform as well as the standard network and cases where linearized wide networks outperform their standard width counterpart.

- As for wider networks the generalization gap closes, in line with NTK theory, we raise the question if this means that the non-wide and wide standard network generalize due to a very different mechanism: feature learning for non-wide networks and effectively employing unlearned random features at larger widths.

- We extend the discussion in previous work of numerical aspects of training the non-wide linearized models by considering the effective rank of the kernel.

## 2 RELATED WORK

The original motivation and prevailing appeal of (finite) deep neural networks is that they are powerful methods to extract statistics and learn features leading to strong performance for down-stream tasks (regime I in Fig. 1) [Lee et al., 2009, Alekseev and Bobe, 2019]. The behavior of neural networks in the highly over-parameterized regime has been extensively studied, too, suggesting minor weight changes from initialization during training (regime II) [Du et al., 2019, Allen-Zhu et al., 2019, Zou et al., 2020]. In the NTK literature, typically, the infinite-width limit for finite-depth neural networks is considered (connection between

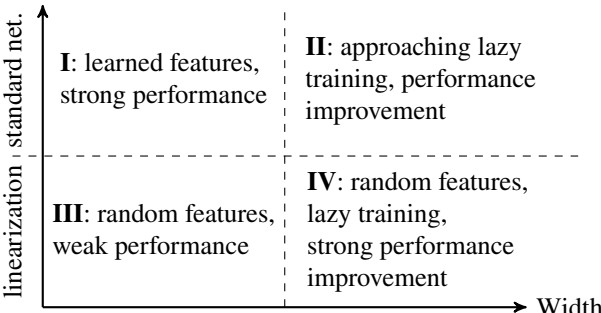

Figure 1: Our results on neural networks exhibit different behavior in different regimes: For wide architectures, standard networks and their linearization become increasingly alike. While the performance of linearized networks benefits substantially from width, standard networks only show small improvements. At usual widths, standard networks and their linearization behave differently due to the relevance of feature learning.

regimes II and IV). In contrast, Deep Equilibrium Models consider the infinite-depth limit at finite width [Bai et al., 2019]. Hanin and Nica [2020] study the NTK for both infinitely wide and deep ReLU networks, showing particular data-dependent features of the resulting NTK in these limits. Focusing on the finite-depth case, there are several studies which compare the finite-width NTK $\Theta$ or infinite-width NTK $\hat{\Theta}$ to their standard network (regimes I and IV) and provide, to some extent, diverging results.

The original work by Jacot et al. [2018] gives experimental results for small synthetic datasets, as well as fully-connected networks trained on MNIST of widths $n = 10^2, 10^3$, and $10^4$, showing good agreement with the infinite-width NTK for the widest network. Lee et al. [2019] extend the original work and show good agreement for small synthetic datasets (of size $\leq 256$) and for a two hidden layer fully-connected network trained with SGD on MNIST. Most interestingly, a wide ResNet trained on CIFAR-10 shows similar behavior, though the non-linearized model appears to have been trained only to below 80% training accuracy, and in the test accuracy a gap seems to develop towards the end of training (see Fig. 7 in their paper). In contrast to that, in Chizat et al. [2019] VGG-11 and ResNet-18 – trained on CIFAR-10 and widened with a scaling factor $\alpha$ for tuning the models into the non-linearized and linearized regimes – exhibit large gaps in test accuracy. They highlight that the decreased training performance of the linearization is due to bad conditioning and effectively low rank of the associated kernel matrix. In their extension to CNNs, Arora et al. [2019] compare CNNs with two to 20 convolutional layers combined with fully-connected or global average pooling output layers to the derived infinite-width convolutional NTK (CNTK), observing large gaps in test accuracy on CIFAR-10.

Our work is most closely related to Lee et al. [2020] and Geiger et al. [2020]. In Lee et al. [2020] an extensive empirical study of neural networks, their linearizations and the infinite-width NTK as well as the Neural Network Gaussian process (NNGP) kernel [Lee et al., 2018, de G. Matthews et al., 2018] is conducted. For fully-connected and simple convolutional architectures, they show cases where NTK can both outperform but also underperform their corresponding networks on CIFAR-10. Importantly, they study the relevance of regularization of the kernels and identify bad conditioning of the kernel as a reason for decreased performance. In line with results by Wei et al. [2020], they showed that $L^2$ regularization (like weight decay) of the kernel is required for good performance in practice, although this breaks the infinite-width correspondence to kernel methods. In contrast to their work, we focus on two more standard but also more extensive CNNs where we increase the widths of the standard and linearized networks explicitly and study their properties with a focus on feature learning. In that regard, our work differs from Geiger et al. [2020] who also study lazy training and feature learning but for fully-connected networks of depth three to five and CNNs with four convolutional layers and in the framework of Chizat et al. [2019] with a scaling factor $\alpha$ controlling the lazy training regime. Another related line of research was conducted by Ortiz-Jiménez et al. [2021] which study linearizations with respect to task complexity defined on the basis of the NTK eigenfunctions as targets. In their evaluation on CIFAR-10, they show that linearization performance can rank learning complexity and show that neural networks do not always outperform their kernel approximations.

Other relevant work includes Seleznova and Kutyniok [2022] which investigates the ordered and chaotic phase phenomena of vanishing and exploding gradients in the context of NTK theory, providing guarantees when the NTK is ill-conditioned (ordered phase) or well-conditioned (chaotic phase and at the border between the two phases). Furthermore, Yang and Hu [2021] note that standard and NTK parameterizations do not lead to representations that learn features in the infinite-width limit and propose an alternative parameterization enabling feature learning in this limit.

## 3 METHOD

We examine two standard ReLU CNNs, LeNet and AlexNet, trained for classification tasks of increasing difficulty. One task is digit recognition in MNIST [LeCun et al., 1998] and modified versions which include random translations of the otherwise centered digits. In addition, we train on CIFAR-10 [Krizhevsky et al., 2009], and a subset of ImageNet [Russakovsky et al., 2015] which contains ten different snake classes (see Supp. Sec. A.1), whereby we deliberately chose similar classes to form a challenging classification task.

In this setup, we study the performance of the standard net-

work and its linearization $f_{\text{lin}}$ (see Eq. (2)) and the effect of increasing the width of the networks, thereby investigating the relationships between regimes I and III as well as III and IV in Fig. 1. This is done by multiplying the number of channels in each convolutional layer and all widths of fully-connected layers by a common factor. Due to GPU memory limitations, we are able to train LeNet and *LinLeNet* up to width factors of 60 and for AlexNet and *LinAlexNet* up to width factors of 4. As the number of parameters increase quadratically in the width, and standard width LeNet and AlexNet have about 60k and 60$m$ parameters, we were hence able to train networks of up to 216$m$ and 960$m$ parameters, respectively.

Our implementation makes use of PyTorch's [Paszke et al., 2019] standard modules for defining and training neural networks with our own custom-made modifications for linearization of the architectures. For LeNet, we adapt the original LeNet-5 architecture [LeCun et al., 1998] to use max-pooling and ReLU activations. For AlexNet, we use the Py-Torch implementation [Krizhevsky, 2014] with 10 outputs rather than 1000 (see below). Despite training for classification, we use the $L^2$ loss with one-hot encoded target vectors. Firstly, with standard cross-entropy loss the networks never converge to exactly zero loss, so the networks must at some point leave the region where the approximation in Eq. (2) is valid, causing some ambiguity in the heuristic. Secondly, the $L^2$ loss allows for an easier and more efficient implementation of the training of the linearized models. We furthermore do not make use of dropout, since it is not clear to us how to model it in the NTK framework (see however Novak et al. [2020]). We find that after optimizing hyperparameters, we can train LeNet and AlexNet to similar train and test performance as with cross-entropy loss without dropout (see Supp. Sec. A.1). We predict the class whose one-hot vector is closest to the output vector, which is equivalent to predicting the argmax of the output layer. We train $f_{\text{lin}}(u, x)$ with SGD in the standard way by optimizing $u$ with gradient updates obtained by

$$\nabla_u \left| f_{\text{lin}}(u, x) - y \right|^2 = 2 \sum_{l=1}^{L} \nabla_w f^l(w_0, x) \times \left( f_{\text{lin}}^l(u, x) - y^l \right). \quad (3)$$

Computing the gradients of the linear model with $L$ outputs requires computing $L$ gradients of the original network per data point, and thus $L$ backward passes, which is computationally intensive if $L$ is large. We therefore train (Lin)AlexNet on the snakes subset of ImageNet consisting of $L = 10$ classes, while we can use full MNIST and CIFAR-10 for (Lin)LeNet.

As our goal is to stay as close as possible to standard neural network training practices, we use SGD with weight decay and momentum. In addition, we use the standard PyTorch weight initialization, which is a variant of Kaiming initializa-

tion [He et al., 2015], rather than the NTK parameterization used in the NTK proofs.

## 4 EXPERIMENTAL RESULTS

In the experiments, five independent reruns of the specified networks for 100 epochs and batch size 32 were performed unless stated otherwise. Hyperparameter search was conducted for each network architecture and its linearization at all widths separately, for learning rates including $\{1, 0.1, 0.01, 0.001\}$ and weight decay values including $5 \times \{10^{-4}, 10^{-5}, 10^{-6}, 10^{-7}, 10^{-8}\}$. The momentum parameter was set to the default value of 0.9. For each rerun, a different fixed random seed was used to ensure that both the standard and linearized models at a particular width are initialized exactly the same and receive the same mini-batches during training. For experiments involving CIFAR-10 and the snakes dataset, the learning rate was decreased by a factor 10 every 30 epochs. Otherwise, we follow the standard pre-processing for standardizing the input images and standard resizing (256 pixels) and center cropping (224 pixels) for ImageNet images. Computations were conducted on Nvidia GeForce Titan X Pascal and Tesla V100 GPUs. For experiments involving LinAlexNet×3 and LinAlexNet×4, we used an Nvidia Quadro RTX 8000 with 48 GB memory due to the increased memory requirement. All displayed results are obtained with single precision. We also carried out all experiments in section 4.1 with double precision, but did not observe any striking differences.

### 4.1 CLASSIFICATION WITH INCREASING FEATURE LEARNING REQUIREMENT

**LeNet trained on MNIST and CIFAR-10:** For LeNet with about 60k parameters, we used width factors ranging from 1 to 60. In all experiments involving MNIST and CIFAR-10, a learning rate of 0.1 and weight decay of $5 \times 10^{-5}$ led to overall best test accuracies.

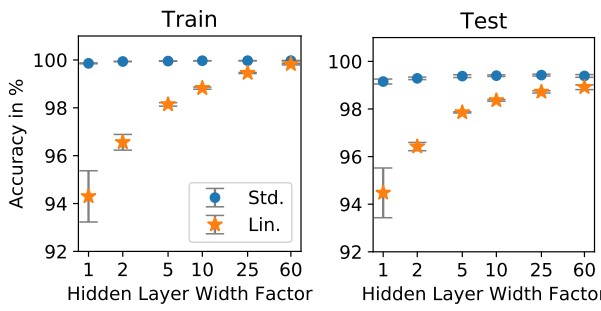

Figure 2: Accuracy of LeNet (•) and LinLeNet (⋆) trained on MNIST at different widths (values in Supp. Table A.2).

The results for MNIST are presented in Fig. 2. For the stan-

dard width, a substantial difference of 4.67 percentage points in (mean) test error between LeNet and LinLeNet is observed. While LeNet does not gain appreciably from increasing the width, LinLeNet does, and the gap shrinks to 0.48 percentage points for width factor 60. Similarly, though not close in a path-wise sense, the statistics of trajectories of output values become more alike with increasing width (shown in Supp. Fig. A.2), indicating a more similar behavior of training dynamics of the linearized and standard models at large width factors.

For factor 1 the linearized model outperforms a logistic regression on normalized MNIST pixels only by a small margin, which achieves about 93% train and 92% test accuracy. The low training accuracy of the linearized models is investigated in more detail in Sec. 4.2 and 4.3.

When increasing the problem difficulty by randomly translating the digits horizontally and vertically, larger gaps in test (and train) accuracy are observed which also decrease with width. For factors 1 and 60, we observe 27.65 and 3.91 percentage points difference in test error. The full results are illustrated in Fig. 3 for translations up to 7 pixels (i.e. up to a quarter of the image size).

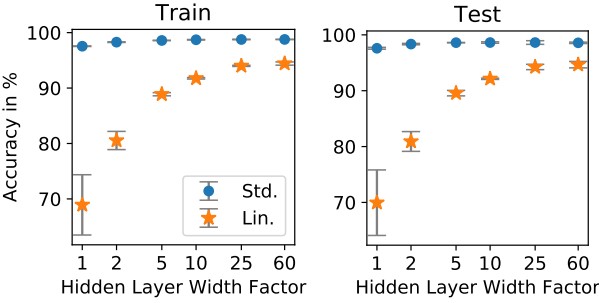

Figure 3: Accuracy of LeNet (•) and LinLeNet (⋆) trained on MNIST at different widths (values in Supp. Table A.2). Digits were shifted randomly by up to 7 pixels.

When training on the more challenging CIFAR-10 dataset even larger gaps are observed, as shown in Fig. 4. For the standard width, a difference of 20.22 percentage points in test error between LeNet and LinLeNet is observed. This shrinks to a smaller but still appreciable gap of 13.17 percentage points at width factor 60. Interestingly, LinLeNet×60 outperforms standard width LeNet×1 in both training and test error (gray dashed line).

**AlexNet trained on snakes dataset:** For AlexNet with about 60m parameters, width factors 1, 2, 3, and 4 were used and the networks were trained on the ten-class snakes subset of ImageNet (see Supp. Sec. A.1). For the linearized networks, a learning rate of 1 and weight decay of $5 \times 10^{-7}$ provided the best test errors. For the standard networks, however, a learning rate of 0.1 and weight decay of $5 \times 10^{-6}$ lead to best test performance. In addition, we trained the

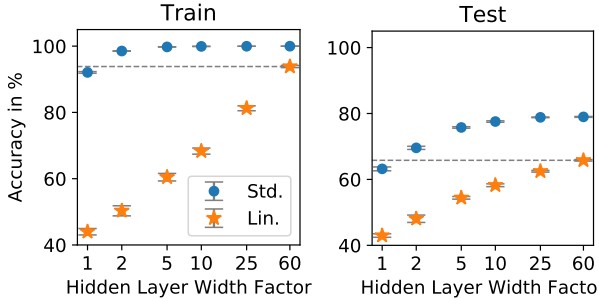

Figure 4: Accuracy of LeNet (•) and LinLeNet (⋆) trained on CIFAR-10 at different widths (values in Supp. Table A.2).

linearized networks with these hyperparameters settings, too, for comparison.

Figure 5 summarizes the findings, which fall in line with the observed trend for LeNet but give even larger gaps in test error. Trained with the same hyperparameters as their non-linearized counterparts, the gaps in test error between standard AlexNet and LinAlexNet are more than 20 percentage points at all considered widths. For the optimal hyperparameters in the linearized setting, the generalization gap shrinks only slightly to 20.4 and 18.56 percentage points for widths 1 and 4. While increasing the width has little impact on train and test error of AlexNet, for LinAlexNet the test error shows a slight decrease and the training error a strong decrease with width.

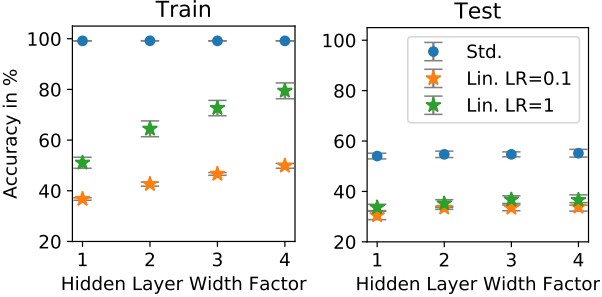

Figure 5: Accuracy of AlexNet (•), LinAlexNet with learning rate 0.1 (⋆) and learning rate 1 (⋆) trained on the snakes dataset at different widths (values in Supp. Table A.1).

These results show that, at standard width or small width expansion factors, the random feature models given by the linearized networks perform poorly compared to their standard network counterpart or the random feature models of wider linearized networks. With increasing problem difficulty, the increasing gap between linearized and standard LeNet and AlexNet suggests that at standard widths significant feature learning is taking place in the standard (non-linearized) model. But with increased over-parameterization, these gaps indeed shrink as predicted by NTK theory. The way the

gap shrinks is through a dramatic improvement in performance of the linearized networks with width, while standard networks are less affected in their performance by width. However, as theory proves that the wide standard trained networks behave as random feature models, we hypothesize that the small improvements in accuracy of standard networks with width might be hiding a significant transition is the underlying *reason* for their good performance, namely from feature learning for the non-wide networks to utilizing non-learned random features that apparently provide a good inductive bias for the tasks at hand for the wider networks (both linearized and non-linearized).

## 4.2 NUMERICAL ASPECTS

The low training accuracy of the non-wide linearized models in the previous experiments raise the question of whether they are well-trained at all. Fitting the linearized model with $m$ data points $x \in \mathbb{R}^d$ is effectively solving the linear system

$$y - f(w_0, x) = \nabla_w f(w_0, x) \cdot u \tag{4}$$

for weights $u \in \mathbb{R}^p$ and target $y \in \mathbb{R}^L$, where $p$ is the number of parameters of the original model and the rows of the matrix $\nabla_w f(w_0, x)$ are the gradients of each output of the network at data point $x$. With $X \in \mathbb{R}^{m \times d}$, the matrix $\nabla_w f(w_0, X) \in \mathbb{R}^{n \times p}$ has $n = 10 \times m$ rows since one must fit each of the $L = 10$ outputs for each data point. LeNet at width factors 1 and 2 has roughly $p_1 = 60k$ and $p_2 = 240k$ parameters, respectively. Thus, the matrix $\nabla_w f(w_0, X)$ cannot have full rank when fitting a dataset of size $m = 50k$ (CIFAR-10) or $m = 60k$ (MNIST), i.e. $p_1, p_2 < 10 \times m$, making it impossible to fit arbitrary targets. Moreover, even for wider networks it appears that matrix $\nabla_w f(w_0, X)$ remains effectively of low rank.

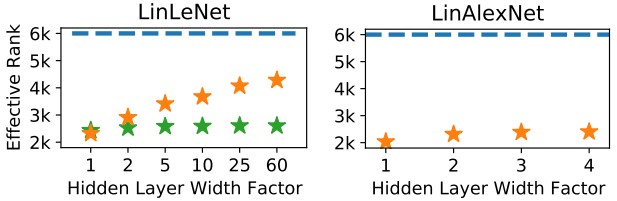

Figure 6: Effective rank of matrix $\nabla_w f(w_0, X)$ in LinLeNet and LinAlexNet for 600 data samples of the MNIST (standard in ★ and shifted in ★) and snakes datasets, respectively, with full rank 6000.

We quantify this by computing the *effective rank* [Roy and Vetterli, 2007] which takes the distribution of singular values into consideration and can be viewed as the exponential entropy of normalized singular values (see Supp. Sec. A.3). For computational reasons, we consider $m = 600$ data samples and the corresponding $6000 \times 6000$ kernel matrix $\nabla_w f(w_0, X) \nabla_w f(w_0, X)^\top$ for each width factor. For

LinLeNet and considering MNIST samples, these effective ranks are much lower than the number of rows, i.e. 6000, and increase with width factor as illustrated in Fig. 6 (left). A similar but less pronounced improvement in effective rank with width is obtained for MNIST samples with additional random translation of up to 7 pixels. Although AlexNet×1 with about 60m parameters is well in the over-parameterized regime for $m = 13k$ datapoints and thus $n = 130k$ rows in matrix $\nabla_w f(w_0, X)$, we still observe large gaps in training accuracy. As for LinLeNet, we show for 600 examples in Fig. 6 (right) that the effective ranks at all widths are significantly lower than the number of rows of $\nabla_w f(w_0, X)$ and increase with width (marginally).

In Fig. 7, the distribution of singular values $\sigma$ of the kernel matrix is shown (see Supp. Fig. A.3 for LinAlexNet). We observe that increasing the width effectively increases the smallest (non-vanishing) singular values of matrix $\nabla_w f(w_0, X)$ and generally leads to a lower condition number (i.e. ratio $\sigma_{\max}/\sigma_{\min}$), for this matrix, thereby improving numerical properties.

We suspect that, in order to perfectly fit the training data, one needs to fit $u$ also in a subspace with very small singular values, making it difficult to achieve close to 100% train accuracy with SGD with non-infinitesimal step sizes.

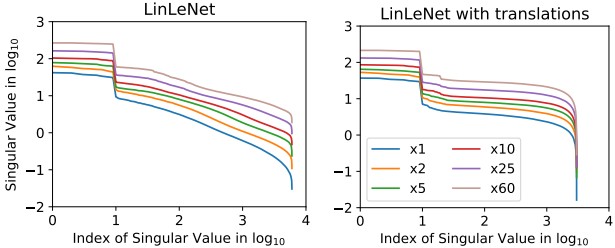

Figure 7: Singular value distribution of LinLeNet for 600 samples of MNIST and MNIST digits randomly shifted by up to 7 pixels. See Supp. Fig. A.3 for LinAlexNet.

## 4.3 BINARY CLASSIFICATION ON MNIST

In order to study these numerical aspects in more detail, we take a closer look at the solution of the linear system in Eq. (4). In particular, we examine if the multiclass setting might be the cause for numerical stability issues due to having multiple outputs (the different classes) for a single input, potentially leading to e.g. collinearity of rows in the matrix. Therefore, we consider binary classification with one output and train to classify a digit as 0 or not 0. Qualitatively similar results were obtained for other target classes. In the following, we solve the one-vs-rest classification task with the same least-squares objective in three ways: by training LeNet with SGD, by training LinLeNet with SGD, and by using a standard solver for linear systems. The presented

results are obtained from single runs of the respective model with a fixed random seed.

**Solving the linear system with SGD:** The training is performed in the same manner as before, but with a learning rate of 0.01 and for 200 epochs. Tables 1 and 2 summarize the binary classification results with target class 0 for standard MNIST and MNIST with up to 7 pixels translation. The qualitative behavior with SGD training follows the same trend as in Fig.s 2 and 3 for the multiclass results (for this reason an illustration is omitted). As before, by including translations of up to 7 pixels of the digits, we observe a drop in accuracies which is particularly pronounced for the linearized setting.

Table 1: Accuracy of LeNet, LinLeNet and the solver on binary MNIST (0 vs. not 0) at different widths.

| | | ×1 | ×2 | ×5 | ×10 | ×25 | ×60 |
|---|---|---|---|---|---|---|---|
| **Test** | Solver | 97.8 | 99.86 | 99.89 | – | – | – |
| | Lin. | 99.61 | 99.75 | 99.81 | 99.85 | 99.85 | 99.89 |
| | LeNet | 99.89 | 99.89 | 99.88 | 99.89 | 99.89 | 99.88 |
| **Train** | Solver | 100 | 100 | 100 | – | – | – |
| | Lin. | 99.42 | 99.74 | 99.88 | 99.97 | 99.9983 | 100 |
| | LeNet | 100 | 100 | 100 | 100 | 100 | 100 |

In comparison to the harder multiclass task, the gap in training accuracy between LeNet and LinLeNet is greatly reduced but persists for the less wide networks, especially for LinLeNet×1 in the translated MNIST task. While training the standard network consistently leads to perfect training accuracy in the standard MNIST setting, it is not possible to achieve 100% training accuracy when solving the linear system in Eq. (4) with SGD, in most cases. However, from a width factor of 5 on, we observe for LinLeNet in the standard MNIST task that the linearized networks start agreeing (up to the second decimal place) with the results of the corresponding LeNet. In particular, LinLeNet×60 matches the train and test results of LeNet at all considered widths, which is in agreement with NTK theory.

Table 2: Accuracy of LeNet, LinLeNet and the solver on binary MNIST (0 vs. not 0) at different widths. Input digits were randomly shifted by up to 7 pixels.

| | | ×1 | ×2 | ×5 | ×10 | ×25 | ×60 |
|---|---|---|---|---|---|---|---|
| **Test** | Solver | 86.05 | 98.7 | 99.17 | – | – | – |
| | Lin. | 95.48 | 98.51 | 98.97 | 99.31 | 99.51 | 99.55 |
| | LeNet | 99.72 | 99.80 | 99.82 | 99.77 | 99.82 | 99.86 |
| **Train** | Solver | 100 | 100 | 100 | – | – | – |
| | Lin. | 95.42 | 98.29 | 98.90 | 99.25 | 99.49 | 99.61 |
| | LeNet | 99.77 | 99.83 | 99.86 | 99.90 | 99.92 | 99.90 |

**Solving the linear system with a standard solver:** Since SGD is not able to attain high train accuracy for linearized models for all widths, it raises the question whether a different algorithm can, and if so, what its generalization properties are for the tasks at hand. An advantage of the binary classification setting is that we can directly solve the linear system in Eq. (4) for $u$ for width multipliers 1, 2, and 5, as the amount of memory required to store the entire matrix $\nabla_w f(w_0, x)$ in memory is reduced and becomes manageable. Larger widths were not feasible for us as more than 1 TB of memory is required even for binary classification, without taking additional memory requirements for the computation into account. We make use of the SciPy least-squares solver which utilizes the highly optimized LAPACK library [Anderson et al., 1999]. The results are included in Tables 1 and 2.

Interestingly, the solver attains perfect training accuracy in all considered cases, but at the cost of a diminished test accuracy for LinLeNet×1 in standard MNIST (see Table 1) and, particularly, in translated MNIST (see Table 2), indicating overfitting of the solver solution. Apparently, the implicit regularization of the SGD solution significantly improves generalization for these widths, while precluding a perfect train accuracy. For LinLeNets of larger widths, an improved generalization is attained which we view to match the SGD results to a reasonable degree (considering fluctuations in the second decimals place as in the multiclass results, see Supp. Table A.2). In the standard MNIST task, the attained solver solutions for LinLeNet×2 and LinLeNet×5 match the test accuracies of their corresponding standard LeNets at otherwise 100% train accuracy. It should be noted that the solver results were obtained without regularization. Additional regularization should lead to similar results as for LinLeNet×1 trained with SGD, that is higher generalization and lower training accuracy.

Therefore, it appears that the observed generalization gaps and poor performance of non-wide linearized models in Sec. 4.1 are not due to poor training optimization. We suspect that moderately wide linearized networks in the multiclass experiments operate in a similar regime as LinLeNet×1 in the binary classification setting.

## 5 CONCLUSION

Motivated by conflicting results in NTK literature, we studied two classical convolutional neural networks, LeNet and AlexNet, and their corresponding linearizations at different widths and increasing difficulty of classification tasks. We investigated four regimes of different behavior in neural networks (see Fig. 1) which complement previous results on lazy training [Chizat et al., 2019] and random feature models [Lee et al., 2019, 2020] summarized in the following.

Firstly, in agreement with previous results like by Arora

et al. [2019], we observed significant train and test performance gaps between standard width LeNet and AlexNet and their corresponding linearization. By considering different classification tasks of increasing difficulty, we showed that the performance gaps increase accordingly suggesting that richer features need to be learned, which the effectively random feature models LinLeNet and LinAlexNet cannot provide.

Secondly, in agreement with work such as Lee et al. [2019], Jacot et al. [2018], we showed, however, that width improves the performance of linearized networks significantly. We hypothesize that the comparatively minor improvements in performance of standard networks might hide a transition from feature learning to utilizing random features at moderate widths. This might be related to previous results suggesting that the intermediate representations of networks of increasing width become increasingly alike to each other and to the representation in the large width limit [Kornblith et al., 2019].

Thirdly, we showed that numerical aspects like the effective rank (see Fig. 6) and distribution of singular values (see Fig. 7 and A.3) of the feature mapping $\nabla_w f(w_0, X)$ have a role in explaining low training accuracy of SGD trained non-wide linearized models. Increasing width appears to remedy these numerical issues of the associated kernel.

In summary, our investigation is based on the finite-width NTK at initialization and explores deviations, as described above, but also convergence to NTK theory. In particular, we show agreement in performance of standard networks and their linearization as well as an instance where a wide LinLeNet(×60) outperforms its standard width LeNet(×1) on CIFAR-10.

Our study highlights the need to study theoretical descriptions of neural network generalization beyond the finite-width NTK at initialization, for instance by considering time-dependent NTK [Huang and Yau, 2019, Jacot et al., 2018] for finite-width networks (see e.g. Fort et al. [2020]) or by further developing the various proposed *mean-field* theories [Chizat and Bach, 2018, Hu et al., 2019, Javanmard et al., 2019, Mei et al., 2019, Nguyen, 2019, Rotskoff et al., 2019]. Additionally, it highlights the need to study the nature of the potential transition to effectively random features at moderate widths in standard neural network training.

## Author Contributions

D. Belius and M. Samarin jointly conceived the idea and wrote the paper. M. Samarin wrote the code, performed the experiments, created the figures and revised the paper. V. Roth participated in the discussions and the refinement of the results.

## Acknowledgements

We are grateful to Levent Sagun, Peter Zaspel, and Ivan Dokmanić for enlightening discussions regarding the results presented in this article. M.S. would like to thank the Swiss National Science Foundation for supporting the research with grant 167333 as part of the Swiss National Research Programme NRP 75 "Big Data". Calculations were performed at sciCORE (http://scicore.unibas.ch/) scientific computing core facility at University of Basel and Amazon Web Services (AWS).

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
