# OpenReview forum: "Feature Learning and Random Features in Standard Finite-Width Convolutional Neural Networks: An Empirical Study"
_auai.org/UAI/2022/Conference — UAI 2022 Poster_

### Official Review · Reviewer_Saht · 2022-03-28

**Q2(1) Originality/Novelty:** 2
**Q2(2) Significance/Impact:** 3
**Q2(3) Correctness/Technical Quality:** 4
**Q2(6) Clarity Of Writing:** 4
**Q6 Overall Score:** 8
**Q8 Confidence In Your Score:** 4

**Q1 Summary And Contributions:**

- This paper conducts an empirical analysis of LeNet/AlexNet and their NTK for increasing width factors  on MNIST, CIFAR10 and an ImageNet subset.
- The paper reinforces the fact |linearized model acc - the original network acc| reduces with width, while the original network's acc remains the same.
- The paper also notes that the SGD-trained linearized models do not have 100% train acc. So they run a solver that gets 100% accuracy, and show that surprisingly, this worsens the test accuracy.

**Q2 Assessment Of The Paper:**

More detailed information regarding each of these aspects is given below:

**Q2(4) Quality Of Experiments (Optional):**

4: Excellent: The experimental evaluation is comprehensive and the results are compelling.

**Q2(5) Reproducibility:**

3: Good: Key resources (e.g., proofs, code, data) are available and key details (e.g., proofs, experimental setup) are sufficiently well-described for competent researchers to confidently reproduce the main results.

**Q3 Main Strengths:**

- The paper looks at a key emerging debate in understanding how neural networks learn and generalize. The paper sheds much required clarity on this debate by reinforcing some existing facts and also by carefully analyzing some aspects of this.
- The results are reported on standard datasets and on reasonable architectures that a typical ML reader will appreciate.
- The paper is extremely well-written.
- There is an excellent coverage of the canonical work in the NTK space, along with a careful and detailed description of the most closely related work.
- The experiments are well-described and well-motivated. I enjoyed read the paper overall.

**Q4 Main Weakness:**

- When it comes to conceptual novelty, it seems to me that the paper doesn't make a significantly new point. However, the specific experimental results themselves are new and helps provide additional clarity on an important, ambiguous phenomenon.
- On this note, I think the paper could be clearer about its difference from Lee et al., 2020. Is the difference simply a change in architecture?

**Q5 Detailed Comments To The Authors:**

Minor typo:
NKT theory -> NTK theory

**Q7 Justification For Your Score:**

The paper provides experiments shedding light on and reinforcing findings in an important and ongoing area of research in understanding NN training. Although I'd not say that the paper is strong in terms of conceptual novelty, the community would find the paper valuable in gaining more clarity, and as supporting evidence for facts regarding NTK training.

**Q9 Complying With Reviewing Instructions:**

1: Yes.

---

### Official Review · Reviewer_extB · 2022-04-04

**Q2(1) Originality/Novelty:** 3
**Q2(2) Significance/Impact:** 3
**Q2(3) Correctness/Technical Quality:** 2
**Q2(6) Clarity Of Writing:** 4
**Q6 Overall Score:** 7
**Q8 Confidence In Your Score:** 2

**Q1 Summary And Contributions:**

The authors investigate the apparently contradictory results in the literature regarding the empirical performance of networks linearized according to Neural Tangent Kernel (NTK) theory. They do this by focusing on realistic (convolutional) networks and tasks. Their main contribution is to show that the discrepancy between theory and measurement can be understood in terms of different network regimes and training challenges.

**Q2 Assessment Of The Paper:**

More detailed information regarding each of these aspects is given below:

**Q2(4) Quality Of Experiments (Optional):**

4: Excellent: The experimental evaluation is comprehensive and the results are compelling.

**Q2(5) Reproducibility:**

3: Good: Key resources (e.g., proofs, code, data) are available and key details (e.g., proofs, experimental setup) are sufficiently well-described for competent researchers to confidently reproduce the main results.

**Q3 Main Strengths:**

The main strength of the submission is that it clearly resolves issues around the practical reality of NTK theory. Their experiments are carefully executed, and show that insufficient width and difficulties in training are both likely explanations for the apparent discrepancies between theory and measurement reported in the literature. Although practical challenges prevent the authors from conclusively showing that these discrepancies will be removed asymptotically, their results are strongly suggestive in this regard. The submission also, in passing, presents a prototypical case where the prior implicit in gradient-based training substantially improves generalization over a more exact optimizer

**Q4 Main Weakness:**

To me, the least convincing part of the paper is the repeated claim to the effect that '[t]he comparatively minor improvements in performance of standard networks [with increasing width] might hide a transition from feature learning to utilizing random features at moderate widths.' The fact that such small changes in test-set performance are observed in two out of three tasks would, on the contrary, suggest that standard CNNs do not switch into an alternative mode at moderate width. This is not unambiguously contradictory with NTK theory, I believe - different modes of generalization could co-exist even at great width values.

**Q5 Detailed Comments To The Authors:**

I found a few small typos in the submission:
- Par before eqn (2): 'function give' should be 'function given'
- Last par of Sec 4,1: 'underling' should be 'underlying'
- In conclusion, correct 'we showed that numerically aspects like the effective rank ...'

Also, I don't believe it is standard practice to italicize metric multipliers (e.g. the k in 60k).

**Q7 Justification For Your Score:**

NTK is a fairly prominent approach to modeling the generalization of DNNs; the authors do a good job of clarifying its empirical status. I do not believe that my main concern is central to any of their claims: if they agree with my assessment, the corresponding claim could be deleted from the paper without detracting from the rest of the contribution.

**Q9 Complying With Reviewing Instructions:**

1: Yes.

---

### Official Review · Reviewer_FCyR · 2022-04-11

**Q2(1) Originality/Novelty:** 2
**Q2(2) Significance/Impact:** 2
**Q2(3) Correctness/Technical Quality:** 3
**Q2(6) Clarity Of Writing:** 2
**Q6 Overall Score:** 6
**Q8 Confidence In Your Score:** 3

**Q1 Summary And Contributions:**

A few years ago, it was theorized that infinitely wide NNs essentially behave as linear random feature models (“linearization” of the NN). This has since been tested empirically in various architectures with diverging results. Here, the authors perform such an empirical study on two CNNs, LeNet & AlexNet, confirming that both outperform their linearized counterpart with a smaller “gap” for wider networks/easier tasks.  They also discuss numerical aspects of training linearized models.

**Q10 Ethical Concerns (Optional):**

No.

**Q2 Assessment Of The Paper:**

More detailed information regarding each of these aspects is given below:

**Q2(4) Quality Of Experiments (Optional):**

2: Fair: The experimental evaluation is weak: important baselines are missing, or the results do not adequately support the main claims.

**Q2(5) Reproducibility:**

3: Good: Key resources (e.g., proofs, code, data) are available and key details (e.g., proofs, experimental setup) are sufficiently well-described for competent researchers to confidently reproduce the main results.

**Q3 Main Strengths:**

A main strength of the paper is the technical rigor of the study design. The authors robustly compare standard and linearized architectures by systematically varying/checking multiple factors:
2)	2 different architectures
3)	classification task difficulty
4)	Including an extra check with a binary classification
5)	In-depth characterization of numerical issues, going in depth to explain the low training accuracy of the linearized networks.
This means the paper is not just a list of results but allows the authors to go in more depth and explain/contextualize their results.


**Q4 Main Weakness:**

1)	Novelty: The introduction does not clearly define a knowledge gap. It mentions that the paper “complements previous work in that direction”, but how exactly it does so remains unclear even in section 2. See Q5 for details.
2)	Significance: Moreover, it is unclear *why* this is important for the field (see also Q5 for details on what was unclear).
3)	Not al conclusions seem to match the results (or perhaps I do not understand correctly what to look for). For example, it is stated in the conclusion (but also other places): “Secondly, in agreement … width improves the performance of linearized networks significantly”. But this seems to be contradicted by some results:
-	LeNet on CIFAR test accuracy gap does decrease but not nearly as much as some of the others (Fig4)
-	AlexNet (fig 5) test accuracy gap barely changes at all with width, certainly not “significantly” (if this refers to effect size rather than its standard statistical meaning).

Can you please explain this discrepancy?


**Q5 Detailed Comments To The Authors:**

Major comments/questions:
1)	Novelty is completely clear (see also Q4). In particular, is the novelty in:
-	The type/complexity of architecture considered (i.e. is this the first time any study is done on AlexNet/LeNet?)
-	The type of analysis performed (i.e. were these networks proviously studied, but not at this level of detail), or
-	A combination thereof (i.e. both the networks studied and the in-depth analysis new to the field).
2)	Significance/impact is not completely clear (see also Q5).
-	what (if any) are the practical implications of wide nets converging to linear models, and why is it important to know this for AlexNet/LeNet specifically?
-	Is this of practical importance for people using these networks, or is this of more theoretical interest? The latter is also fine, but what is then the added theoretical value of confirming this phenomenon in these networks?
-	The discussion starts with “Motivated by conflicting results in NTK literature…” – but does the current paper offer a resolution, or “just” add to the body of evidence?
3)	Contradiction conclusions-results: see Q4.
4)	P4 “Similarly, though not close in a path-wise sense, …” – but isn’t this contradicting the theory in the introduction, which says “when training these models … steps, then the training trajectories will stay close”? Please comment on this discrepancy and whether or not it is important for interpreting the results.
5)	P8 Last paragraph – “highlights the need to study … NTK at initialization”. It was not clear to me what this means or how this follows from the work. Could you expand on this?

More minor comments (no need to respond):

Some phrases in the manuscript somewhat vague and could benefit from a more concrete rephrasing, e.g.:
6.	Introduction: “which extended important insights about the connection of Gaussian processes and neural networks” -> what kind of insights?

7.	Introduction: “we complement previous work…” -> complement how? I get that space is limited and this is partially the purpose of sec2 “related work”, but the introduction should already state what specific knowledge gap the paper is trying to address.

8.	Conclusion: “which complement previous results on lazy training and random feature models” -> complement how?

9.	Conclusion: “motivated by conflicting results in NTK literature” – this motivation was not clear to me until I read it here; it might help to emphasize this more in the introduction.

Very minor comments (no response needed):
-	P7 just before conclusion: muTLiclass -> muLTiclass
-	P8 “numerically aspects” -> “numerical aspects”



**Q7 Justification For Your Score:**

Technically the study is sufficiently rigorous to be published. My main concerns are novelty and impact, which are currently difficult for non-expert readers to assess. This might be solved if the authors can explain in the rebuttal or can (slightly) rewrite the intro to emphasize this more. Another weakness was the discrepancy between the main conclusion and the AlexNet results, but this might again be quite easily fixed by further explanation.

**Q9 Complying With Reviewing Instructions:**

1: Yes.

---

### Official Review · Reviewer_GAAw · 2022-04-16

**Q2(1) Originality/Novelty:** 1
**Q2(2) Significance/Impact:** 2
**Q2(3) Correctness/Technical Quality:** 3
**Q2(6) Clarity Of Writing:** 3
**Q6 Overall Score:** 5
**Q8 Confidence In Your Score:** 3

**Q1 Summary And Contributions:**

The paper provides and empirical comparison of CNNs and their linearisation on a set of small scale image classification tasks (MNIST, CIFAR10, ) with varying levels of difficulty (with respect the number of classes). The paper aims to study the connection between feature learning and using random feature in NNs and linearised NNs in finite and infinite width settings.

**Q2 Assessment Of The Paper:**

More detailed information regarding each of these aspects is given below:

**Q2(4) Quality Of Experiments (Optional):**

2: Fair: The experimental evaluation is weak: important baselines are missing, or the results do not adequately support the main claims.

**Q2(5) Reproducibility:**

4: Excellent: Key resources (e.g., proofs, code, data) are available and key details (e.g., proof sketches, experimental setup) are comprehensively described for competent researchers to confidently and easily reproduce the main results.

**Q3 Main Strengths:**

- Experimental settings and hyper-parameters are explained in good details.
- The paper illustrates the role of factors such as effective rank and distribution of singular values on the performance of  linearised NNs trained with SGD.

**Q4 Main Weakness:**

- The significance of some of the contributions of the paper is not clear: Two of the mentioned contributions of the paper are: (1) Demonstrating empirically that finite-width neural networks, generally, outperform the finite-width linearisation of these architecture; and (2) Presenting cases where linearized NNs perform as well as their standard version. The significance of these as a contribution is not clear to me considering previous work such as [1, 2, 3].

- Some of the arguments of the paper are not well supported: It is not clear to me how the larger gap in performance when the classification task is harder (in terms of number of classes) relates to the fact that finite-width NNs perform feature learning while their linearization can not, and how the experiments provided in the paper lead to the conclusion that NNs transition from learning features to using non-learned random features in the over-parametrised regime. Maybe the authors can elaborate a bit more on this.



**Q5 Detailed Comments To The Authors:**

The idea, that NNs transition from learning features to using non-learned random features in the over-parametrised regime is potentially interesting but it requires further analysis to show this beyond speculation. I’d be curious for example to see if the instances of an NN with different width scaling factors, but similar performances are similar with respect to representational similarity metrics such as CKA. To extend this comment a bit further, I think to argue NNs and their linearised version become similar beyond their performance as the width increases, we need to have more extensive analysis comparing them from different aspects. Most importantly this idea that large width causes neural networks to lose the ability to learn features has been challenged by [4], and if the authors believe and can demonstrate otherwise, I think it’s worth discussing it.

Additionally, it would be nicer if you can elaborate a bit more extensively/clearly how your works relates or builds on top of some existing papers on the topic such as [1,2,3,5].

[1] Ghorbani, Behrooz, et al. "When do neural networks outperform kernel methods?." Advances in Neural Information Processing Systems 33 (2020): 14820-14830.

[2]  Refinetti, Maria, et al. "Classifying high-dimensional Gaussian mixtures: Where kernel methods fail and neural networks succeed." International Conference on Machine Learning. PMLR, 2021.

[3] Fort, Stanislav, et al. "Deep learning versus kernel learning: an empirical study of loss landscape geometry and the time evolution of the neural tangent kernel." Advances in Neural Information Processing Systems 33 (2020): 5850-5861.

[4] Yang, Greg, and Edward J. Hu. "Tensor programs iv: Feature learning in infinite-width neural networks." International Conference on Machine Learning. PMLR, 2021.
APA

[5] Ortiz-Jiménez, Guillermo, Seyed-Mohsen Moosavi-Dezfooli, and Pascal Frossard. "What can linearized neural networks actually say about generalization?." Advances in Neural Information Processing Systems 34 (2021).

**Q7 Justification For Your Score:**

Given the extent of the experiments (only small scale experiments), and not much analysis I am not sure if the paper provides any new insights on the topic. There are some interesting points raised in the paper, that if supported by further analysis and investigation can make the paper stronger.

**Q9 Complying With Reviewing Instructions:**

1: Yes.

---

### Decision · Program_Chairs · 2022-05-15

**Decision:**

Accept (Poster)

**Comment:**

Meta Review: This paper provides a comprehensive experimental study on feature Learning in neural networks and learning with random features in linearized networks. While there is still some reservation about the significance of the contribution, all the reviewers agree that this work is valuable, and is a good empirical work to backup existing theoretical work in both NTK and mean-field regime.  While the experiments are on classification problems with cross-entropy loss, the NTK works the authors cited and discussed are using square loss (e.g., Du et al. ). Therefore, it is necessary to cite and discuss the NTK work for classification problems with cross entropy loss [1] [2] in the introduction and related work sections, to close the gap between the theory and experiments. Please address this point and also the suggestions by the reviewers carefully in the camera ready.

[1] Zou et al., Gradient descent optimizes over- parameterized deep relu networks. Machine Learning, 109(3):467–492, 2020.
[2] Allen-Zhu et al., A convergence theory for deep learning via over- parameterization. arXiv preprint arXiv:1811.03962, 2018.